# Vitamin D Supplementation Practices among Multiple Sclerosis Patients and Professionals

**DOI:** 10.3390/jcm11247278

**Published:** 2022-12-08

**Authors:** Weronika Galus, Anna Walawska-Hrycek, Michalina Rzepka, Ewa Krzystanek

**Affiliations:** 1Department of Neurology, Faculty of Medical Sciences in Katowice, Medical University of Silesia, 40-752 Katowice, Poland; 2Department of Neurology, Faculty of Health Sciences in Katowice, Medical University of Silesia, 40-635 Katowice, Poland

**Keywords:** multiple sclerosis, vitamin D, supplementation

## Abstract

Vitamin D serum level increase is associated with a reduction in clinical relapse rate, gadolinium-enhancing lesions, new or enlarging T2 lesions and new active lesions in the MRI in MS patients. However, current RCTs assessing the vitamin D supplementation therapeutic effect in MS provide inconclusive results. Experts recommend vitamin D measurements and implementations among patients with MS. This article discusses an observational study, performed without any intervention to evaluate the vitamin D status and practices among MS patients and professionals in the clinical setting. A total of 139 patients with MS treated by disease-modifying therapy were recruited and fulfilled the standardized questionnaire assessing the vitamin D supplementation practices and vitamin D level influencing factors such as education, insolation, smoking, obesity and current treatment. The collected data were then compared to the patients’ vitamin D serum levels available in medical records at the baseline and after 12 months of observation. Professionals’ practices and recommendations were also assessed. A total of 74.1% patients confirmed vitamin D supplementation, and all patients were administered cholecalciferol, taken orally. However, only 43.69% of the patients achieved an optimal vitamin D concentration (30–50 ng/mL). The lack of a doctor’s recommendation was the most frequent reason for the absence of vitamin D supplementation. The most often recommended vitamin D daily dose was 4000 IU. There was no adverse effect of supplementation observed. Vitamin D status in patients with MS is currently better than in the general population, but still, a significant percentage of patients do not implement vitamin D.

## 1. Introduction

Multiple sclerosis (MS) is one of the most common inflammatory demyelinating and degenerative diseases of the central nervous system (CNS) among young adults. Age-adjusted prevalence and incidence in Poland is high, ranging from 6.6 to 131.2, even in 244.9/100,000 inhabitants, respectively, in 2019 [1]. Approximately 2.8 million people are estimated to live with MS globally [2]. Unknown factors trigger excessive autoimmune attacks leading to myelin and axon damage [3]. Four disease courses have been identified: clinically isolated syndrome (CIS), relapsing remitting MS (RRMS), primary progressive MS (PPMS) and secondary progressive MS (SPMS) [4]. The diagnosis is based on McDonald criteria (revised in 2017) that fulfil dissemination in time and space of lesions in the CNS [5]. The course of MS is extremely variable, but its progression causes disability, which can be assessed through the Expanded Disability Status Score (EDSS) [6]. MS is an incurable disease, but disease-modifying therapy (DMT) can slow its development [7]. Both genetic and environmental factors contribute to developing MS. Vitamin D deficiency has been established as one of the most relevant risk factors [8]. Newer genome-wide association studies provided robust evidence that low serum 25(OH)D concentration is a cause of MS, independently of other recognized risk factors [9].

The term vitamin D refers to a family of compounds that are essential to maintain bone and teeth health, modulate the immune response and function as a neurosteroid [10]. Cholecalciferol (vitamin D3) is a natural form of vitamin D included in food as well as supplemented and produced in skin upon irradiation. This synthesis is the most important source of vitamin D and depends on latitude and season, which is not achievable in Central Europe, including Poland, between late September and early April [11]. Application of a sunscreen with an SPF of 30 reduces the vitamin D skin synthesis by 95–98% [12]. Cholecalciferol is hydroxylated in the kidneys and liver to active metabolite calcitriol (1,25-dihydroxyvitamin D, 1,25(OH)2D3). 1,25(OH)2D3 acts by vitamin D receptor (VDR) [13]. 1,25(OH2)D3 blocks the progression of murine experimental autoimmune encephalomyelitis (model of multiple sclerosis) [14,15].

In patients with MS, vitamin D supplementations reduce the production of proinflammatory cytokine [16] and elevate the levels of anti-inflammatory cytokines such as transforming growth factor-beta (TGF-ß) and interleukin-10 (IL-10) [17,18]. Inverse associations between serum 25(OH)D levels and cerebrospinal fluid neurofilament light (CSF-NFL) levels, a marker for axonal injury in patients with MS, were also revealed [19]. Vitamin D treatment may also have potentially remyelinating effects by promoting the proliferation of oligodendrocyte precursor cells (OPC) [20]. A meta-analysis by Martinez-Lapiscina et al., including thirteen eligible studies with 3498 patients, revealed that an increase in serum 25(OH)D levels was associated with a reduction in clinical relapse rate, gadolinium-enhancing lesions, new or enlarging T2 lesions and new active lesions in magnetic resonance imaging (MRI) in MS patients [21]. Furthermore, a meta-analysis by Moosazadeh et al., involving fourteen studies with 2817 participants with MS, showed a significant negative correlation between 25(OH)D level and disability expressed by EDSS [22]. Nevertheless, current randomized clinical trials (RCTs) assessing the therapeutic effect of vitamin D supplementation in MS patients provided inconclusive results. Jagannath et al. in a 2018 Cochrane meta-analysis (including 12 RCTs comprising 464 patients in treatment and 469 patients in control group) disclosed no benefit of vitamin D administration on recurrence of relapses, EDSS and MRI lesions [23]. McLaughlin et al. also presented comparable results [24]. The newest metanalysis by Zheng et al., including nine eligible RCTs with 356 patients in treatment and 362 patients in the control group, disclosed that management of vitamin D did not affect the EDSS outcomes and relapses during the research [25]. Hanaei et al. [26], Doostri-Irani et al. [27] and Zheng et al. [28] presented analogous results.

Polish endocrine experts’ vitamin D supplementation guidelines recommend vitamin D supplementation in patients with MS. If the insolation instructions are not fulfilled (sunbathing with uncovered forearms and legs for at least 15 min between 10.00 a.m. and 3.00 p.m., without sunscreen from May to September), a supplementation of 800–2000 IU/day is recommended over a year. The recommendation also includes an evaluation of vitamin D serum level [29].

Based on these current data, an analysis of vitamin D status and supplementation in MS patients is warranted. The aims of this study were:Evaluation of the vitamin D status among MS patients in the clinical setting.Analysis of supplementation in clinical practice (percentage of supplementation; measurement of vitamin D concentration, dosages and period of supplementations; effectiveness expressed by vitamin D serum level; patients’ and professionals’ practices).Evaluation of factors influencing the effectiveness of supplementation in MS patients (patients’ knowledge, professionals’ recommendations, insolation, body mass index (BMI), smoking, comorbidities).Evaluation of vitamin D implementation safety and intoxications.

## 2. Materials and Methods

Between October 2018 and April 2022, 147 patients with MS, treated in the Department of Neurology of the Medical University of Silesia in Katowice, Poland, were recruited. All patients provided informed consent to participate in the study. Inclusion criteria were:Diagnosed with MS (RRMS, SPMS, PPMS) according to the 2010 or 2017 revised McDonald criteria;Aged > 18 years;EDSS ≤ 6; andDMT treatment (I-line: interferon beta, glatiramer acetate, dimethyl fumarate, teriflunomide; II-line: natalizumab fingolimod, ocrelizumab, alemtuzumab).

Exclusion criteria were as follows:Relapse in the past 4 weeks;Cognitive function impairment at dementia stage;EDSS > 6;Other treatment (mitoxantrone, cyclofosfamde);Pregnancy, breast-feeding;Acute or chronic renal failure; andHypercalcemia in medical history.

Participants were required to fill out the standardized questionnaire about demographic (age, gender, education, height, weight, skin phototype) and MS clinical features. The knowledge about vitamin D and its supplementation recommendation was also assessed. Patients who supplemented vitamin D presented researchers with characteristics of their vitamin D usage (dosages, time, reasons of supplementation, regularity, doctor’s recommendations and administrations, vitamin D measurements); the rest were asked about the reason for a lack of supplementation. Vitamin D level influencing features such as insolation, sunscreen usage, diet, smoking, comorbidities and steroid use were also evaluated. Obtained data were compared to the clinical characteristics collected from the available medical records as well as to the vitamin D levels at baseline and after 12 months of observation. Vitamin D serum levels in this group were standardly measured during annual visits. Vitamin D serum levels were assessed by 25(OH)D3 expressed in ng/mL. The vitamin D concentrations were applied according to Polish endocrine experts’ recommendation, as follows: severe deficiency (0–10 ng/mL), deficiency (>10–20 ng/mL), suboptimal (>20–30 ng/mL), optimal (>30–50 ng/mL), high (>50–100 ng/mL) and toxic (>100 ng/mL) [30].

The study was conducted in accordance with applicable regulations. The opinion of the Bioethics Committee of Medical University of Silesia was obtained (KNW/0022/KB/135/19).

Statistical analyses were performed in Statistica 13.2., StatSoft Polska, Krakow, Poland. The Shapiro–Wilk test was used to check the data normality. In case of non-normal data distribution, nonparametric tests were applied. The Mann–Whitney U test was used to assess statistical significance between distinctive groups. *p* < 0.05 was considered statistically significant.

## 3. Results

In total, 139 participants were recruited, due to incorrect questionnaire completion by eight candidates. The study group included 104 women and 35 men; the median age was 40.5 years. The patients primarily had RRMS (n = 137); the rest had PPMS (n = 2). Patients were administered I-line (n = 118) and II-line (n = 21) DMT. The characteristics of the study group are shown in Table 1.

### 3.1. Patient Practices

A total of 74.1% patients confirmed vitamin D supplementation, but only 65% had been supplementing for more than 6 months. All patients were administered cholecalciferol, taken orally. Vitamin D was taken daily by 71.84% of patients. Approximately 43.69% of patients of this group achieved optimal vitamin D concentrations, whereas insufficient (lower than optimal) vitamin D serum levels were recorded in 36.83%, including deficiency and severe deficiency in 11.65% patients with supplementation. High vitamin D levels were attained in 15.52%, and toxic levels ranged from 114 ng/mL to 164 ng/mL in 3.88% of patients at the baseline (Table 2).

The following daily doses were used by patients with supplementation: 2000 IU (38.8%), 4000 IU (35.92%), 1000 IU (12.62%), <500 IU (6.8%) and >4000 IU (3.88%). Over 70% of patients declared taking vitamin D every day. Supplementation throughout the year was conducted by 59.22% of the patients and only in autumn and winter by 32.04% of the patients. Mean vitamin D serum levels according to vitamin D daily dose are presented in Table 3.

In the group without supplementation, the optimal vitamin D serum level was recorded in 25% of patients, whereas insufficient (lower than optimal) levels were recorded in 72.22%, with deficiency in 33.33% and severe deficiency in 16.67% of patients at the baseline. A high level was noted only in one person.

At the baseline, the mean vitamin D concentration in the study group was 35.07 ng/mL (SD ± 24.27), 39.80 ng/mL in the group with vitamin D supplementation and 21.55 ng/mL in the group without supplementation, with statistically significant differences between groups (*p* = 0.000001).

After 12 months of observation, the change in vitamin D serum levels was statistically significant in patients with supplementations (*p* = 0.000073), whereas there was no statistically significant change in patients without supplementation (Table 4). The patients with toxic vitamin D serum levels (n = 4) were not involved in the 12-month observation due to the necessity to reduce their vitamin D serum levels.

### 3.2. Patients’ Reasons for Supplementation

A doctor’s recommendation was the most frequently mentioned reason for vitamin D supplementation, noted in 64% patients. Meanwhile, 36.9% of patients indicated their own decision to improve their health and 21.55% of patients expressed concern about vitamin D deficiency. Patients’ reasons for supplementation are presented in Table 5.

Over one-quarter of the study group denied vitamin D supplementation. The lack of a doctor’s recommendation was the most frequent reason for vitamin D supplementation absence (41.67%). Low compliance and polypharmacy were the next most frequent reasons. Patients’ reasons for lack of supplementation are presented in Table 6.

### 3.3. Doctors’ Practices and Recommendations

The main MS doctors recommended vitamin D supplementation in 55.4% of treated patients and explained supplementation’s legitimacy to 48.9% of patients. The recommendations and explanations resulted in supplementation in 98.53% of patients. Only one person did not start supplementation despite doctors’ references. A higher mean vitamin D serum level was recorded in this group (39.43 ng/mL) compared to the group without a doctor’s recommendation (29.77 ng/mL), independently from vitamin D usage (*p* = 0.003).

After 12 months of observation, there were statistically significant changes in vitamin D serum levels (*p* = 0.000042) among patients with doctors’ recommendations and explanations independently from supplementation (Figure 1).

In available medical records vitamin D supplementation, written recommendation was found in 46.04% of cases. The following daily doses were recommended: 4000 IU (50%), 2000 IU (17.18%) and 1000 IU (10.93%). Recommendations without specific doses constituted 20.31% of cases. Vitamin D supplementation withdrawal and control measurement of vitamin D serum level recommendation was noted in one patient. Furthermore, 63.31% of patients in the study group confirmed vitamin D serum level measurements, once in 52.27% and regularly in 47.73% of patients.

### 3.4. Disease-Modifying Therapy and Vitamin D Serum Level

There were no statistically significant differences between mean vitamin D serum levels in patients treated with I-line and II-line DMT in the study group. Regardless of supplementation, the mean vitamin D serum level was 36.07 ng/mL in patients treated with I-line DMT and 29.46 ng/mL in those treated with II-line DMT. No statistically significant differences were noted.

### 3.5. Insolation, Smoking, Body Mass Index, Education and Vitamin D Serum Level

Among patients who fulfilled insolation guidelines (38.85%), the mean vitamin D serum level was 37.18 ng/mL, while in patients with insufficient sunbathing, it equaled 33.71 ng/mL (*p* = 0.02). Analogous results concerning insolation were noted in patients both with supplementation (*p* = 0.03) and without supplementation (*p* = 0.04). There were statistically significant differences between the analyzed groups. Moreover, worse vitamin D status was associated with higher BMI, but there were statistically significant differences only in patients without supplementation (*p* = 0.03). Although worse vitamin D status was noted in smokers, there were no statistically significant differences regarding smoking.

No correlations between vitamin D serum levels and the educational status of patients were observed. Almost all patients confirmed having knowledge about vitamin D. The most common information sources were, in descending order, healthcare providers, including doctors (61.87%), media (55.40%) and family and friends (41.73%). Levels of vitamin D knowledge were analyzed among the study group. Higher mean vitamin D serum levels were noted in patients who provided more correct answers; however, there were no statistically significant differences.

## 4. Discussion

The study provided data about vitamin D status and supplementation among MS patients as well as patients’ and professionals’ actions in the clinical setting. Despite numerous systematic reviews and meta-analyses concerning vitamin D level associations and the benefits of supplementation among patients with MS, studies evaluating the current vitamin D usage among MS patients are scarce.

### 4.1. Vitamin D Status in MS Patients

Regardless supplementation, optimal or higher vitamin D serum levels (>30 ng/mL) were reported in 53.94% of patients in the study group, whereas in the general populations of the Polish cities, vitamin D serum levels were less than optimal in 89.9% of people [31]. This could be a result of a higher percentage of vitamin D supplementation among MS patients than the general population.

### 4.2. Vitamin D Supplementation

In this study, 103 out of 139 patients with MS were administered vitamin D (74.1%), more than in previous studies. Masullo et al. reported 19 subjects declaring vitamin D supplementation among 36 patients with MS (52.78%) [32], whereas Pape K. et al. noted 111 out of 185 individuals with MS supplementing vitamin D (60.0%) [33]. The results indicate that supplementation of vitamin D is effective with 12 months of supplementation.

### 4.3. Professional Practices

Professionals recommended vitamin D supplementation in 55.4% of patients, generally 4000 IU/day, and only 63.31% of patients have had vitamin D serum level tested before the study. The recommended vitamin D daily doses were sufficient, but professionals should quantify supplementation and regular vitamin D serum level measurements in every MS patient, in accordance with Polish guidelines. Furthermore, Scandinavian experts advocate for vitamin D administration in MS patients at least in autumn and winter (800 IU/day) or, alternatively, regular vitamin D serum level measurements with proper supplementation [34]. Other authors recommend testing vitamin D serum levels in people with MS, and suggest empirical replacement with 4000 IU/day without additional calcium [35]. Moreover, doses up to 4000 IU/day are recommended even for pregnant women with MS [36].

### 4.4. Effectiveness of Supplementation

Approximately 63.09% of patients with vitamin D supplementations achieved optimal or higher vitamin D serum levels; the rest had insufficient levels despite the supplementation. The results of this study show that doctors’ recommendations and explanations are remarkable factors in influencing the effectiveness of vitamin D supplementation. Other authors indicate that appropriate justification of treatment results in better adherence in MS patients with DMT [37]. Furthermore, the relationship between professionals and MS patients contributes greatly to the treatment process [38].

In the study group, a positive correlation between sufficient insolation and the vitamin D serum levels was noted in patients with and without supplementation. There was no statistically significant evidence regarding the influence of smoking and education. Despite the trends favoring an inverse correlation between BMI and vitamin D status, statistically significant differences were observed only in the group without supplementation.

### 4.5. Vitamin D Supplementation Safety and Toxicity

Toxic vitamin D serum levels ranging from 114 to 164 ng/mL were noted in four patients in the study group (3.88% of patients in the group with supplementation). Three patients used doses greater than or equal to 4000 IU/day, and one patient used 1000 IU/day. Nevertheless, neither adverse effects nor laboratory deviations were observed at the baseline and after 12 months of observation. Patients were recommended to reduce vitamin D doses and have serum levels controlled. Fargoso et al. reported 21 cases of immediate DMT withdrawal and vitamin D administrations in high doses ranging from 8000 IU/day to 150,000 IU/day, resulting in worsening neurological status and severe adverse effects [39]. Alarming non-conventional treatment protocol for MS with high doses of vitamin D (cumulative 78,000,000 IU, mean 130,000 IU/day) may lead to serious intoxication and acute kidney injury [40]. Serious adverse events or intoxications are rarely reported in current randomized clinical trials with vitamin D supplementation up to 20,000/day [41,42,43]. Intoxication was not observed in the study group due to predominant daily doses of up to 4000 IU. Moderate dose administrations and regular vitamin D serum level measurements are the best clinical practices for MS patients. We also suggest that it is essential to inform patients about the elevated risk of non-conventional treatment.

### 4.6. Study Strengths and Limitations

This study was focused on observation with no intervention in order to evaluate the vitamin D status and implementation in the clinical setting. Therefore, we included patients at the beginning stages of the disease, as well as with progressed MS, both RRMS and PPMS, and with almost all types of DMT. The study was conducted in the Department of Neurology of the Medical University of Silesia, where other vitamin D studies among patients with MS were held, so the percentage of patients with MS with vitamin D supplementation may be higher than at other medical centers. However, the studied group was varied and included patients from many cities and villages from Silesia and other regions of southern Poland.

## 5. Conclusions

Vitamin D status in patients with MS is currently better than in the general population, but still, a significant percentage of patients do not supplement with vitamin D. According to current guidelines and experts’ recommendations, there are certain arrangements that should be put into practice regarding vitamin D supplementation commencement and amelioration. Doctors’ recommendations and explanations play a crucial role in supplementation effectiveness among MS patients. Sufficient insolation is associated with higher vitamin D serum levels, whereas smoking and obesity are associated with lower vitamin D status. Vitamin D supplementations in moderate doses (2000 IU–4000 IU/day), supported by regular vitamin D concentration measurements in patients with MS, are safe and guarantee appropriate vitamin D status.

## Figures and Tables

**Figure 1 jcm-11-07278-f001:**
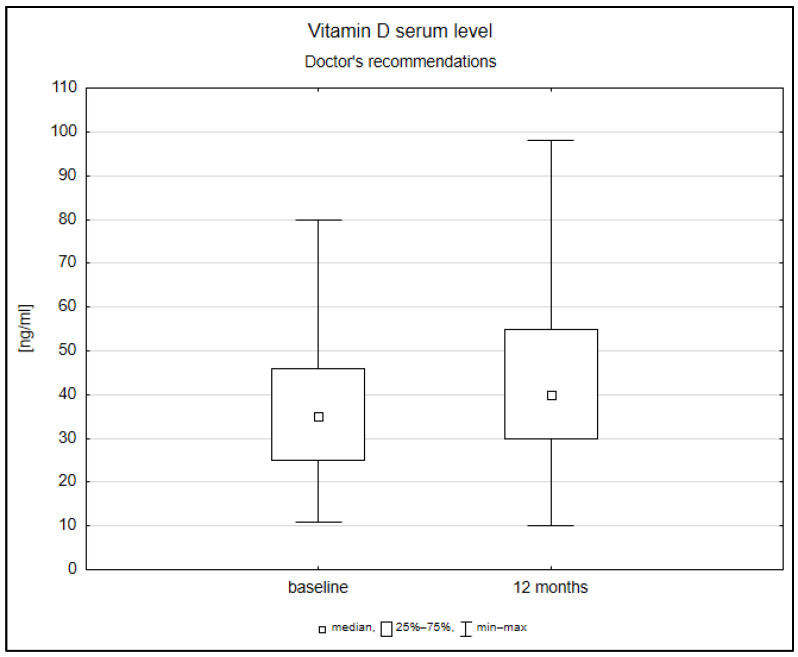
Box blot of change in vitamin D serum levels after 12 months of observation in patients with a doctor’s recommendation for vitamin D supplementation.

**Table 1 jcm-11-07278-t001:** Demographic and clinical characteristics of the study group.

Characteristics	Results (N = 139)
Men/women (n, %)	35 (25%)/104 (75%)
Age (years, mean ± SD)	40.5 ± 10.3
Body mass index (mean ± SD)	25 ± 4.5
RRMS/SPMS (n, %)	137 (98.6%)/2 (1.4%)
Disease duration (years, mean ± SD)	10.2 ± 6.6
Number of relapses (n, mean ± SD)	4.2 ± 2.7
EDSS (median, min–max)	2.5 (1–6)
DMT (n, %)	139 (100%)
*Interferon-beta*	*25 (18%)*
*Glatiramer acetate*	*11 (7.9%)*
*Dimethyl fumarate*	*63 (45.3%)*
*Teriflunomide*	*19 (13.7%)*
*Fingolimod*	*8 (5.8%)*
*Natalizumab*	*10 (7.2%)*
*Ocrelizumab*	*2 (1.4%)*
*Alemtuzumab*	*1 (0.7%)*
Sufficient insolation (n, %)	54 (38.9%)
High vitamin D intake in diet (n, %)	38 (27.3%)
Smoking (n, %)	26 (18.7%)

**Table 2 jcm-11-07278-t002:** Vitamin D serum levels among studied group at the baseline.

Vitamin D Concentration		Studied Group N = 139		Patients with Supplementation N = 104		Patients without Supplementation N = 36	
Serum Level	(ng/mL)	(n)	(%)	(n)	(%)	(n)	(%)
Severe deficiency	<10	7	5.04	1	0.97	6	16.67
Deficiency	>10–20	23	16.55	11	10.68	12	33.33
Suboptimal	>20–30	34	24.46	26	25.24	8	22.22
Optimal	>30–50	54	38.84	45	43.69	9	25
High	>50–100	17	12.23	16	15.52	1	2.78
Toxic	>100	4	2.87	4	3.88	0	0

**Table 3 jcm-11-07278-t003:** Vitamin D serum levels among MS patients according to vitamin D supplementation daily dose.

Vitamin D Dose (IU/Day)	(n)	(%)	Mean Vitamin D Serum Level ± SD (ng/mL)
>4000	4	3.88	87.00 ± 64.05
4000	37	35.92	45.41 ± 25.02
2000	40	38.83	32.26 ± 13.05
1000	13	12.62	40.18 ± 30.62
500	7	6.8%	27.71 ± 13.89

**Table 4 jcm-11-07278-t004:** Median values of 25(OH)D serum levels at baseline and after 12 months of observation among patients with MS.

Vitamin D Serum Level	N	Baseline	After 12 Months	Change	*p* *
Patients with vitamin D supplementation	99	32.70 (8.00, 80.00)	38.00 (12.00, 98.00)	4.90 (−35.10, 47.00)	0.000073
Patients without vitamin D supplementation	36	18.60 (5.00, 54.13)	21.00 (5.00, 58.900	1.6 (−36.13, 44.46)	0.189088

Note: Values are median with minimum and maximum in brackets. * Wilcoxon’s test.

**Table 5 jcm-11-07278-t005:** MS patients’ reasons for vitamin D supplementation.

Reason for Supplementation	(n)	(%)
Doctor’s recommendation	66	64
My own decision to improve health	38	36.9
Concern about vitamin D deficiency	22	21.4
Family’s/friends’ support	5	4.8
Dietetic’s recommendation	4	3.9
Current guidelines	2	1.9
Others	6	5.8

**Table 6 jcm-11-07278-t006:** MS patients’ reasons for lack of vitamin D supplementation.

Reason for Lack of Supplementation	(n)	(%)
Lack of a doctor’s recommendation	15	41.7
Low compliance	11	30.6
Polypharmacy	9	25
Vitamin D supplementation has no benefits	5	13.9
Frequent insolation	4	11.1
Supplementation is unnecessary	3	8.3
High costs of vitamin D preparations	1	2.8

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
