# Peer review of "Vitamin D Supplementation Practices among Multiple Sclerosis Patients and Professionals"

_jcm, 2022, doi:10.3390/jcm11247278_

Round 1

Reviewer 1 Report

The notion that this is a "real world" descriptive study is somewhat misleading, as this is a single-center project involving an MS center so it doesn't really represent the varied practice patterns of multiple practices.

The data on Vit D levels is purely descriptive; it would be helpful to know durations of treatment (if available) and change in Vit D levels over time (if available). If these data are obtained through chart review, these data should be largely discoverable.

Perhaps the most interesting aspect of the study are the results of the patient surveys about reasons for taking or not taking Vit D. The impact of the physician recommendation is notable.

The paper is overly long for the material presented. Much of the data could be more concisely consolidated into fewer tables or figures. Especially for the unremarkable results, these could be succinctly summarized in text rather than occupying an entire table.

The paper is weakened in part by need for extensive English language revision. Improving this and narrowing the focus of the paper should be completed in order to be considered for publication.

Author Response

Manuscript ID#: jcm-2002208

Manuscript title: Vitamin D supplementation practices among multiple sclerosis patients and professionals.

Responses to the Reviewer’s comments:

Thank you very much for your time and effort in reviewing the following article and for the opportunity to submit a revised draft of our manuscript “Vitamin D supplementation practices among multiple sclerosis patients and professionals” for publication in Journal of Clinical Medicine.

Please note, that in our responses we are referring to the page numbers, sections, and paragraphs in the final version of the manuscript. Together with the final manuscript, we are providing tracked-changes version utilizing MS Word Track Changes functionality. New text is marked with the underline and deleted text is marked with the single strike-through.

Here is, in blue, a point-by-point response to the Reviewer’s comments and concerns. Original text coming from the review is provided in italics.

Reviewers #1 Comments to the Author:

Comment #1 The notion that this is a "real world" descriptive study is somewhat misleading, as this is a single-center project involving an MS center so it doesn't really represent the varied practice patterns of multiple practices.

Answer: We agree. The term “real world” could be misleading. We suggest “clinical settings” to emphasis that it was observational study without any interventions.

Comment #2 The data on Vit D levels is purely descriptive; it would be helpful to know durations of treatment (if available) and change in Vit D levels over time (if available). If these data are obtained through chart review, these data should be largely discoverable.

Answer: Thank you for this comment. The questionnaire involved the duration of treatment, we included that “only 65% have been supplementing for longer than 6 months” among patients with supplementation (Section 3.1.). The control vitamin D serum levels after 12 months of observations are also available in patients’ medical records (vitamin D serum level is standardly measured during annual visits). According to your suggestion, we added this data and performed the Wilcoxon’s test to assess the statistically significant changes between vitamin D serum level (median, minimum and maximum) at the baseline and after 12 months of observations among patients with supplementation and patients without supplementations. The results were presented in Table 5.

Comment #3 Perhaps the most interesting aspect of the study are the results of the patient surveys about reasons for taking or not taking Vit D. The impact of the physician recommendation is notable.

Answer: Thank you for this comment. The impact of the physicians’ recommendation is notable. The Wilcoxon’s test revealed statistically significant changes in vitamin D levels over time (p=0,000042) in the study group at the baseline and after 12 months of observations. The results were present in Figure 1 (box plot). We also discoursed it in Discussion (Section 4.4) and compared with studies concerning adherence in MS patient treated with Disease Modifying Therapy, because there is no available data according vitamin D supplementation.

Comment #4 The paper is overly long for the material presented. Much of the data could be more concisely consolidated into fewer tables or figures. Especially for the unremarkable results, these could be succinctly summarized in text rather than occupying an entire table.

Answer: We agree. We decided to delete two tables showing association between vitamin D serum level and DMT as well as the BMI (Table 7 and Table 8 in original manuscript). We also combine the Section 3.4 and 3.5 (Insolation, Smoking, Body Mass Index and Vitamin D Serum Level and Education and Vitamin D Knowledge) and reduce the information that was unremarkable.

Comment #5 The paper is weakened in part by need for extensive English language revision. Improving this and narrowing the focus of the paper should be completed in order to be considered for publication.

Answer: Thank you for this comment. We have had our original manuscript checked by a native English-speaking colleague for appropriate English and edited where necessary. In order to narrow the focus of the paper, we highlighted remarkable results, especially physicians’ recommendations.

Reviewer 2 Report

The authors present an observational study without intervention to evaluate the vitamin D status in MS patients under real-world conditions. This is of importance as on the one hand, vitamin D insufficiency can worsen the course of MS, on the other hand, sometimes patients are ingesting toxic doses of vitamin D. In some cases, this even led to acute kidney injury. It is therefore interesting to have data on the vitamin D use, dosage and serum level in this patient group. Especially clear dose recommendations are currently lacking. The data presented as well as the manuscript appears sound and well written. The limitation is that the PPMS group comprised of only 2 patients. It would be interesting to look at the progressive/late stage patient group independently in a follow-up study.

Could the authors conclude with some recommendation to the patients and their physicians? Probably a regular vitamin D serum level check could be recommendend to avoid toxic complications as well as still low serum levels despite supplementation.

Author Response

Manuscript ID#: jcm-2002208

Manuscript title: Vitamin D supplementation practices among multiple sclerosis patients and professionals.

Responses to the Reviewer’s comments:

Thank you very much for your time and effort in reviewing the following article and for the opportunity to submit a revised draft of our manuscript „Vitamin D supplementation practices among multiple sclerosis patients and professionals” for publication in Journal of Clinical Medicine.

Please note, that in our responses we are referring to the page numbers, sections and paragraphs in the final version of the manuscript. Together with the final manuscript, we are providing tracked-changes version utilizing MS Word Track Changes functionality. New text is marked with the underline and deleted text is marked with single strike-through.

Here is, in blue, a point-by-point response to the Reviewer’s comments and concerns. Original text coming from the review is provided in italics.

Reviewers #1 Comments to the Author:

Comment #1 The authors present an observational study without intervention to evaluate the vitamin D status in MS patients under real-world conditions. This is of importance as on the one hand, vitamin D insufficiency can worsen the course of MS, on the other hand, sometimes patients are ingesting toxic doses of vitamin D. In some cases, this even led to acute kidney injury. It is therefore interesting to have data on the vitamin D use, dosage and serum level in this patient group. Especially clear dose recommendations are currently lacking. The data presented as well as the manuscript appears sound and satisfactory written. The limitation is that the PPMS group comprised of only 2 patients. It would be interesting to look at the progressive/late stage patient group independently in a follow-up study.

Answer: Thank you for this comment. We agree that analysis of vitamin D status and supplementation in multiple sclerosis patients is warranted, especially to avoid vitamin D toxicity. Vitamin D serum level increase is also associated with a reduction in clinical and radiological outcomes of multiple sclerosis patients. There are recommendations about vitamin D supplementation for general population and patients in risk of vitamin D deficiency, as well as certain experts’ and professionals’ opinions, but there is still a necessity for definite recommendation for this group of patients. We are pleased that manuscript and presented data are satisfactory. It is also very interesting point to follow-up the vitamin D status among progressive or late-stage patients with MS.

Comment #2 Could the authors conclude with some recommendation to the patients and their physicians? Probably a regular vitamin D serum level check could be recommendend to avoid toxic complications as well as still low serum levels despite supplementation.

Answer: Thank you for this suggestion. According to study results, we recommend vitamin D supplementations in moderate daily doses (2000IU - 4000IU) and regular vitamin D concentrations measurements in patients with MS to guarantee safe and appropriate vitamin D status. We included it in Section Conclusions.
